# Inverse design of stochastic, voxelated thermo-viscoelastic digital materials

Marc Wirth ®[1,2] ✉, Joël N. Chapuis ®[1,2] & Kristina Shea[1]

Polymer material jetting enables the fabrication of voxelated, multi-material structures with material control at the microscale. However, current work often neglects viscoelastic effects and designing voxelated digital materials remains challenging due to the complexity of the vast design freedom and intractability of efficiently modeling macroscale voxel structures. We present an efficient representation of stochastically mixed, voxelated digital materials and develop a generalized viscoelastic temperature-dependent material model to design and simulate digital materials mixed from two constituent polymers. The material model is based on an extended percolation theory considering frequency and temperature. An artificial neural network is trained on the material model to directly estimate target material behavior given arbitrary non-linear, user requirements. The approach is validated using two case studies requiring tailored, non-linear material behavior: a personalized wrist orthosis and a machine damper. These show the newly unlocked possibilities for the design and fabrication of tuned, stochastic digital materials.

Architected materials have transformed material design due to the unique interplay between the architected mesoscale geometry and the resulting macroscopic mechanical properties[1,2]. The resulting design flexibility enables advances in different applications such as high stiffness lattices[3], optical waveguides[4], origami[5], and multi-state structures[6,7]. The realization of these designs is often supported by Additive Manufacturing (AM) due to the processes capability of producing free-form structures[8–11].

Recently, polymer material jetting has pushed architected materials further by enabling multi-material AM by material assignment per voxel on a microscale[10,12,13]. Prior work has extensively explored this process to design metamaterials with features significantly larger than the voxel scale. The resulting meso-scale features are designed as deterministic elements, often optimized for shape morphing[6,14], stiffness tuning[3], and topology control[15]. These design approaches often assume homogenous or deterministic voxel layouts[16], which enable accurate property estimation by simulation at the meso-scale. In contrast, material design on the voxel-scale enables smaller features but is governed by voxel-scale interaction effects of the base materials, causing the mesoscale material behavior to be dominated by microscale material interfaces rather than only by the base material

properties[17]. This effect is often leveraged to create digital materials exhibiting emergent behavior through the stochastic mixing of base materials[18]. These stochastic digital materials enable, for example, the direct tuning of material stiffness[1,8,14], the polymer shape memory effect[19,20], and the design of continuously graded material interfaces[15,21]. While it has been shown that modeling and simulation of stochastic digital materials can be tackled by coarse-graining and homogenization[1,8], most work relies on an assumption of only elastic behavior[8,9,15,19,22–26] or neglects the dependence on a wide temperature range[18,21] and loading rate[17–20,27]. This impedes the use of voxelated polymer material jetting for use cases with more complex mechanical behavior, such as damping structures and applications requiring temperature-dependent stiffness modulation.

Moreover, the combinatorial scaling behavior of voxel-based material allocation makes inverse material design, in this context, intractable when using conventional computational methods. While various data-driven and optimization-based inverse design frameworks have been proposed for mechanical metamaterials, they often target unit cell design or fixed lattices, are often constrained to linear behavior and are limited to shape or stiffness matching of a single state of a single structure[23,28–31]. Additionally, the currently proposed data-

[1]Engineering Design and Computing Laboratory, Department of Mechanical and Process Engineering, ETH Zurich, Zurich, Switzerland. [2]These authors contributed equally: Marc Wirth, Joël N. Chapuis. ✉e-mail: wirthma@ethz.ch

driven approaches rely heavily on understanding the microscale structure for the automated generation of large datasets[29,30,32–36]. For polymer voxel AM, the lack of voxel-level microscale simulation prevents the systematic generation of artificial datasets and calls for empirical-based modeling[3,14,17]. To date, there is no generalizable and scalable inverse material mapping between desired viscoelastic, time-dependent, and temperature-dependent material response and voxel-scale material composition in stochastically mixed, multi-material systems.

In this work, we present a unified modeling and inverse material design framework for stochastically mixed, voxelated digital materials, combining a generalized viscoelastic material model with a data-driven inverse material design framework (see Fig. 1). The material model homogenizes stochastic digital materials of two base materials using an extension of the physics-based de Gennes percolation theory[37] to account for the time-dependent and temperature-dependent stiffness as well as voxel-scale interaction effects. The model is empirically tuned and validated on experimental data. Based on this model, an Artificial Neural Network (ANN) is trained to directly infer voxelated digital materials from user-defined mechanical design requirements. The proposed inverse material design framework is geometry-independent, does not require problem-specific re-training, and enables the direct design of viscoelastic, voxelated digital materials across a range of time scales and temperatures. The approach is validated through two representative case studies. The first case study considers the modeling of a monolithic, voxelated machine damper under cyclic loading, thus showing the simulation of hysteresis and smoothly graded material interfaces that are used to prevent stress concentrations. The second case study investigates the inverse material design of a personalized, multi-material wrist orthosis that can limit wrist deflection and compensate for swelling and unexpected impacts, thus demonstrating consideration of material requirements at multiple time scales.

## Results

### Efficient representation of voxelated digital materials

The representation of voxelated digital materials quickly exceeds computational limits. While a conventional dogbone type specimen can contain distinct regions of hard and soft materials[38], a voxelated specimen can include graded boundaries to prevent stress concentrations and yields a complex material distribution of 25 million individual voxels (Fig. 2a). Further, a close-up inspection of a voxelated, AM material mixed from two polymers reveals a high amount of material smearing on a scale significantly larger than the theoretical

AM resolution of 84 µm (Fig. 2a). Since computational limits and complex intertwining of material interfaces both prevent the simulation of idealized, voxelated digital materials on the microscale, a representation on an intermediate mesoscale using a coarse grid is proposed, effectively separating the actual voxel layout from the mesoscopic behavior of the digital material. More specifically, macroscopic geometry is approximated by grid points on the mesoscale, where each grid point is assigned a designated material (Fig. 2b). For AM, voxels are assigned a base material stochastically based on the defined digital material mixture. The presented representation approach separates the scales of material assignment from the voxel scale by tuning the resolution of the coarse grid to the finest feature size and enables direct material assignment to Finite Element (FE) meshes for simulation when using a homogenized material model on the mesoscale. The coarse grid resolution is thus adapted to the size of the geometric features in the structure, the desired material assignment resolution, and the FE mesh resolution required for simulation (see Section A.2.4 in the supplementary information).

### Material model: stochastic digital materials and de gennes percolation theory

Previous work shows that the homogenized properties of stochastic mixtures of two base materials can be modeled using de Gennes percolation theory[37]. This theory describes how the homogenized digital material behaves based on the volume fractions of the individual base materials. As the volume fraction of material one $\rho_1$ increases, the mixture morphology transitions from dispersed droplets to an interconnected network[18], thus changing the ratio of the effective properties $r$ non-linearly according to Eq. (1).

$$r = \frac{(\rho_1 - \rho_c)^\mu}{(1 - \rho_c)^\mu} \tag{1}$$

The percolation threshold, $\rho_c$, is set to 0.163 based on previous literature[19], validation by visual inspection of different stochastic digital materials and an optimization procedure based on sample II from Fig. 3a (see supplementary information). The percolation coefficient, µ, for two materials with different stiffnesses is evaluated by testing the two base materials, namely a soft material and a rigid material, and five stochastic digital materials (Fig. 3a) using a dynamic mechanical analysis (DMA). The Time-Temperature Superposition Principle[39,40] (TTSP) is assumed and the materials are subsequently modeled using a generalized Maxwell model represented using the Prony series[41,42] in combination with a Williams-Landel-Ferry[40] (WLF)

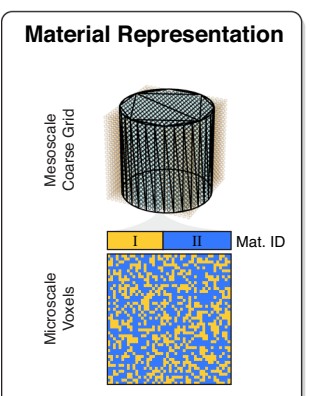
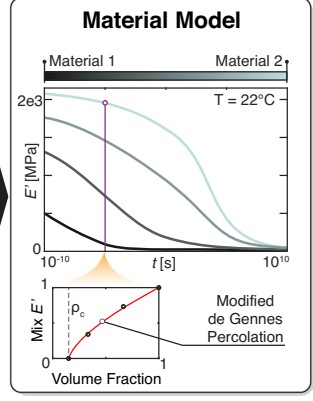
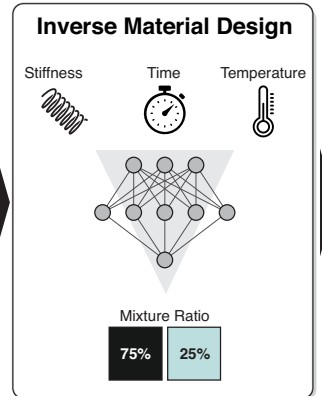
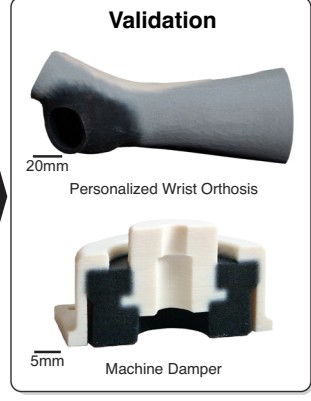

**Fig. 1 | Overview of the presented work.** From left to right: (1) efficient material representation leveraging homogenization; (2) a material model for stochastic digital materials of two polymer base materials is developed using a temperature and frequency-dependent de Gennes[37] percolation theory; (3) an ANN is trained on the material model to estimate the material mixture ratio given required stiffness, time scale and temperature (images provided from left to right by lovemask/ stock.adobe.com, haji/stock.adobe.com, and Ralf's icons/stock.adobe.com); (4) the validity of the proposed modeling approaches is shown by applying them to a personalized wrist orthosis and a machine damper. Source data are provided as a Source Data file.

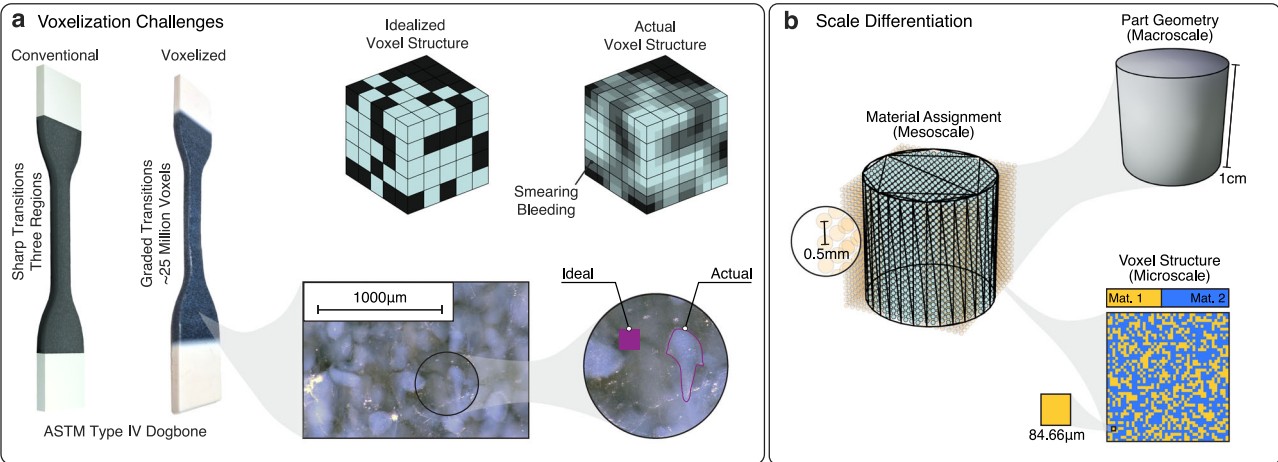

**Fig. 2 | Challenges of voxelated digital materials and the required scale differentiation. a** The two main challenges in representing voxelated, AM material are the extensive number of data points needed to represent macroscale geometries and the stochastic nature of material placement due to manufacturing inaccuracies. This results in smearing between materials and bleeding from one material domain into neighboring domains. **b** For efficient representation of voxelated digital materials, the geometry is approximated as mesoscale grid points, where each grid point is assigned a material mixture. The microscale voxel layout is a stochastic assignment of base materials according to a defined mixture ratio.

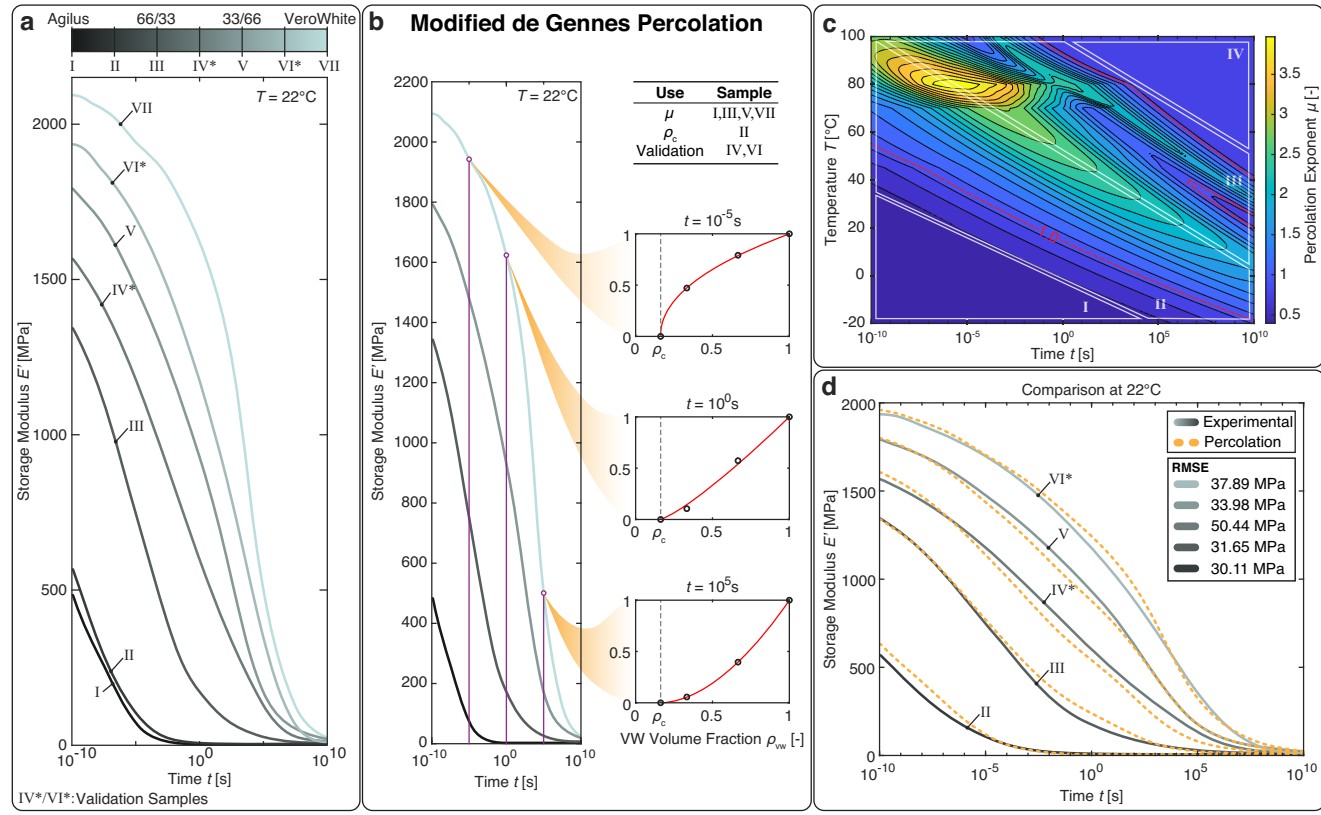

**Fig. 3 | Modified de Gennes percolation model applied to viscoelastic master curves. a** Prony series master curves of all seven experimental samples additively manufactured on the Stratasys J850 Prime using Agilus30Black (AB) and VeroUltraWhite (VW) as base materials. **b** Example de Gennes percolation fitting at three frequencies at 22 °C. **c** Frequency-dependent and temperature-dependent percolation coefficient. **d** Comparison of the mechanical properties of the experimental samples and the predicted master curves created using frequency-dependent percolation. Source data are provided as a Source Data file.

and Arrhenius function[43]. The resulting master curves show a nonlinear dependency between the volumetric mixture ratios and resulting stiffness in the frequency range of $10^{-10}$ Hz and $10^{10}$ Hz. The resulting stiffnesses $E'$ at different frequencies are biased toward the stiffer material at higher frequencies and towards the softer material at low frequencies. This indicates that the percolation coefficient $\mu$ is dependent on frequency and due to the TTSP also on temperature. The percolation coefficient $\mu$ is thus fit per frequency and temperature using four of the seven experimentally determined Prony series curves (Fig. 3b, sample I, III, V, and VII). The percolation threshold, $\rho_c$, is fit using sample II and the final percolation model is validated using samples IV and VI. Additionally, all other mixtures are also compared to the percolation model to assess the quality of the fitting procedure (Fig. 3d).

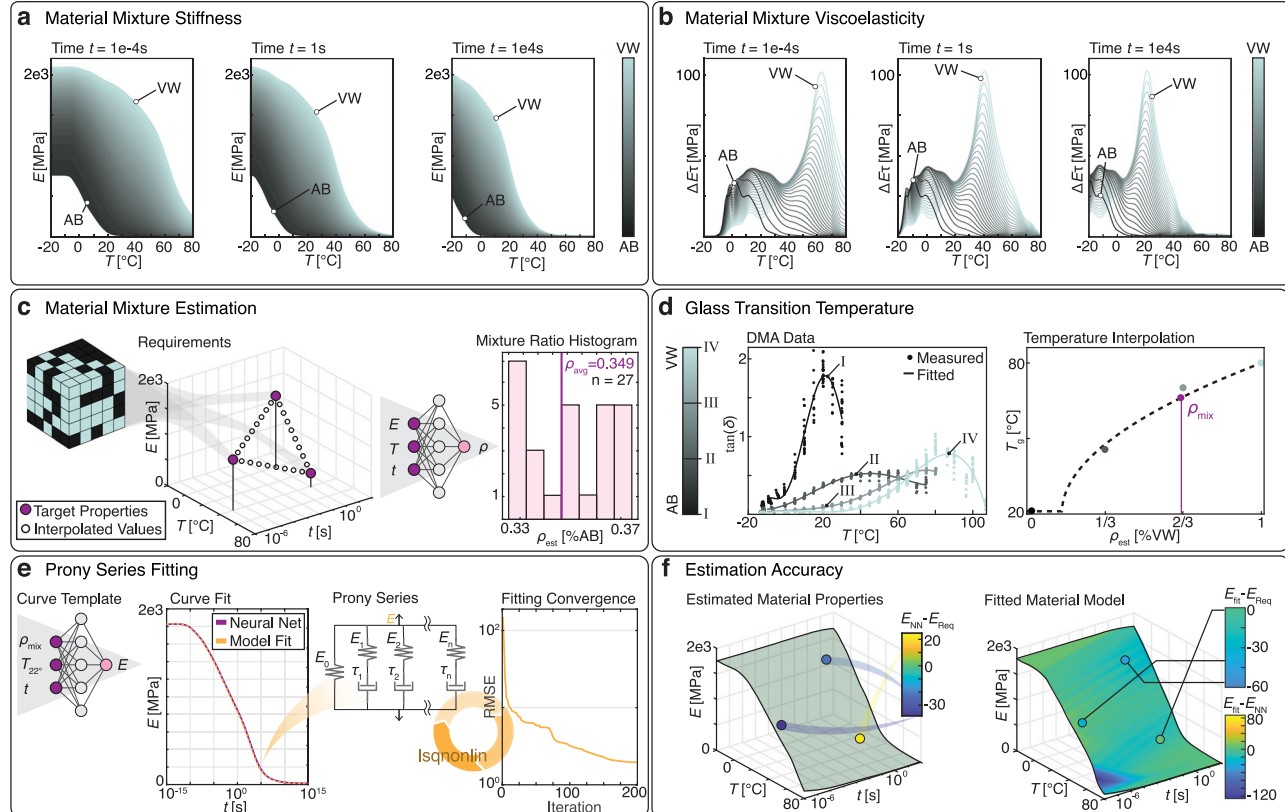

**Fig. 4 | Material estimation procedure for inverse design. a** Material stiffness for three different time scales over all temperatures and material mixtures of a soft material (AB) and a harder material (VW). **b** Material viscoelasticity $\Delta E_\tau$, provided as stiffness decrease over characteristic time, for three different time scales over all temperatures and material mixtures of AB and VW. **c** Estimation of mixture ratio from user requirements. A tuple of requirements is enhanced by interpolated values and fused as input to an ANN that estimates the required mixture ratio. The mean estimate is the target material mixture. **d** The glass transition temperature of the target mixture is interpolated from the experimentally determined values.

Details on the number of samples and testing procedure can be found in Supplementary Table 1. **e** The master curve necessary for Prony series fitting is evaluated at room temperature by an ANN, which is provided with the mixture ratio, the temperature and a series of different times *t*. The Prony series is fit to the estimated master curve (**f**) Left: Accuracy of the example requirement given in (**c**) compared to the stiffness estimation in (**e**). Right: Accuracy of the fit Prony series compared to the stiffness estimation and to the example requirements, respectively. Source data are provided as a Source Data file.

The resulting frequency-dependent and temperature-dependent percolation coefficient is shown in Fig. 3c. At low temperatures and short time spans (Zone I), the percolation coefficient remains nearly constant. This indicates that the two constituent materials exhibit stable behavior, as the Arrhenius function delays the activation of the damper elements in the material model. In Zone II, the percolation coefficient begins to increase with both temperature and time. The coefficient changes smoothly, forming a ridge where its maximum value appears between Zones II and III. In Zone III, the percolation coefficient becomes less well-defined, likely due to significant changes in the stiffness of both constituent materials. As the materials soften, individual Maxwell elements exert a greater influence on the overall behavior, leading to increased local variability. Finally, in Zone IV, the percolation coefficient stabilizes once again. This is because most or all Maxwell elements have fully relaxed, and the coefficient is now governed primarily by the constant long-term stiffness of the constituent materials. The percolation coefficient is generally greater than one for a combination of high frequencies and high temperatures as well as low frequencies and low temperatures (Zone II and III)[37]. In Zones I and IV, the coefficient falls below one. The proposed frequency-dependent version of the de Gennes percolation shows good agreement with the two validation samples that were not included in the fitting process (Fig. 3d), showing a root mean squared error in the Young's modulus across the entire tested frequency range of between 30.11 MPa and 50.44 MPa.

## Material behavior estimation for inverse design

The percolation-based interpolation enables the stiffness estimation of voxelated digital materials at each time scale and temperature. This is used to create a mapping $\mathcal{M}$ between mixture ratio, time scale, temperature and stiffness (Fig. 4a) as well as to the material viscoelasticity, expressed as a stiffness decrease over a characteristic time period (Fig. 4b). While the stiffness is maximal for a higher percentage of the rigid material, the viscoelasticity $\Delta E_\tau$ shows an interesting switching behavior across temperature.

When designing, it is desirable to directly determine the material mixture ratio based on given requirements. In this case, each requirement can be defined as a tuple of target stiffness at a given time scale and temperature. For multiple requirement tuples, the determination of the target mixture becomes ambiguous. Here, the ambiguity is bypassed by training an ANN with a multi-layer perceptron architecture on the mapping $\mathcal{M}$ for both estimating mixture ratios ($ANN_{mix}$) and stiffness values ($ANN_E$) from the remaining three dimensions, respectively and chaining the ANNs into an inverse design framework. Detailed information on the ANN architecture and training can be found in the subsequent methods section and in the supplementary information. In this framework, the user-provided requirement tuples are extended by linearly spaced requirement tuples and the network $ANN_{mix}$ estimates target material mixtures from the extended set of requirement tuples. These estimations are averaged, providing the target mixture ratio $\rho_{mix}$ (Fig. 4c). The target mixture ratio determines

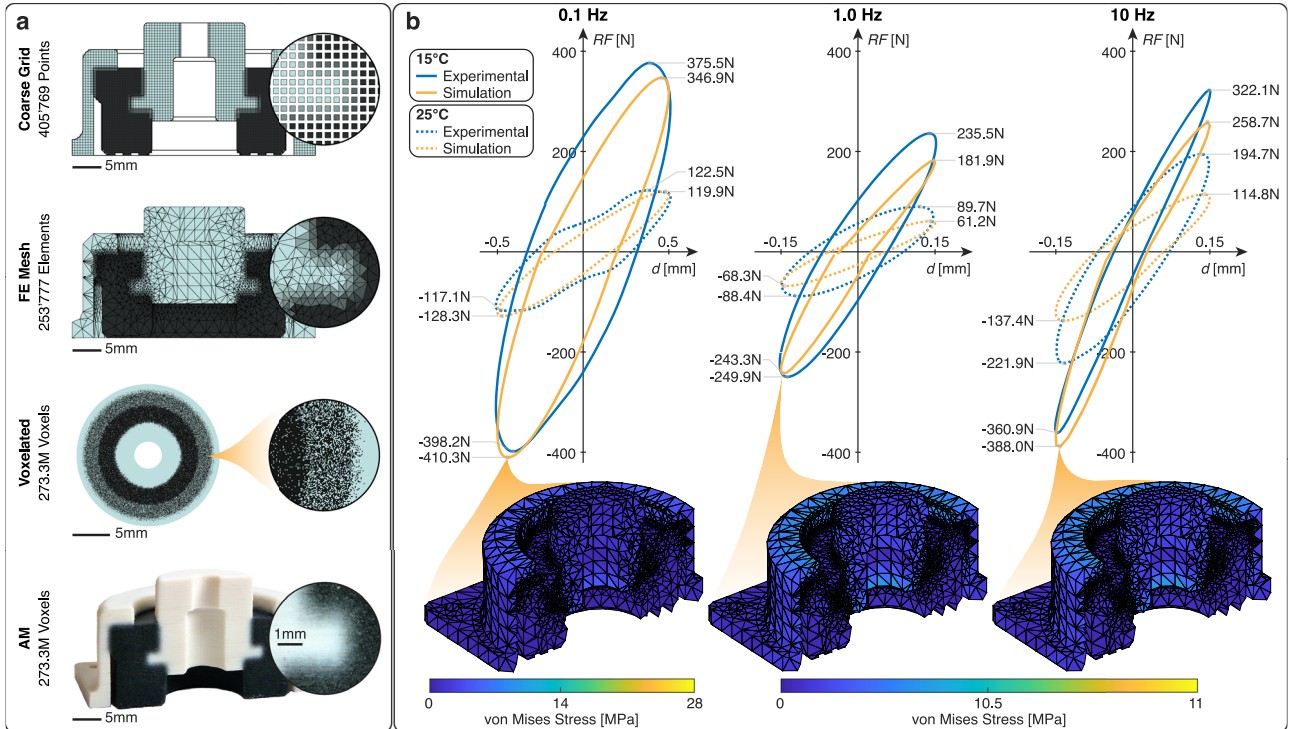

**Fig. 5 | Results related to the machine damper case study. a** Overview of the coarse grid, Finite Element (FE) mesh, voxelated, and AM versions of the machine damper, including relevant entity counts. **b** Comparison of the hysteresis curves of the simulated and experimental AM damper at three different frequencies and two temperatures per frequency. The graphs compare the reaction force *RF* to the displacement *d*. The von Mises stress for the state of maximum compression is shown for the 15 °C simulation for all three frequencies. Source data are provided as a Source Data file.

the glass transition temperature $T_{g,mix}$ by percolation from the experimentally determined $T_{g,test}$ of the tested samples (Fig. 4d). The target mixture ratio is provided as input to the network $ANN_E$, which estimates the material stiffness over the full time range from $10^{-15}$s to $10^{15}$s for the determined mixture at room temperature. The resulting stiffness over time is the master curve for the target mixture. A generalized Maxwell model with hyperelastic stiffness elements is fitted to the master curve using the previously determined $T_{g,mix}$ and an interpolated hyperelastic curve based on experimental data (Fig. 4e). Depending on the diversity of requirement tuples, the degree to which the estimated stiffness of both $ANN_E$ and the fit model fulfill the required stiffness can vary. The agreement between the fit model and the network estimation, however, is a matter of the fitting performance and varies depending on the temperature and time scale (Fig. 4f). The observed differences can be attributed to the Arrhenius shift factor of the experimental curves not being equal, which cannot be depicted using a single Prony series with a single shift factor in the fit model.

Since the inverse material design framework is trained on the theoretical material model, the influence of real-world variability on the estimated properties is evaluated experimentally and analytically (detailed description in supplementary information). Experimental mechanical characterization of identical samples revealed a manufacturing variability of 3% in force response. The analytical analysis of the impact of stochastic base material assignment is conducted using a cube of a given feature size, the number of voxels within the cube determines the effective sample size of the binomial mixing process. Assuming voxel-wise material assignment with a certain probability, the standard deviation of the number of voxels allocated can be quantified. This analysis reveals that the stochastic variability is larger than the manufacturing variability of 3% for features below 1–2 mm, depending on the target mixture ratio. Similar values of around 5 mm can be found in literature[18]. This suggests that the deterministic ANN model behaves accurately for homogenized mesoscale structures.

## Case study: machine damper

A promising example to demonstrate the validity of the proposed material model is the inverse design of a standard machine damper, which is used in a range of applications from soft robotics to aerospace[44,45]. There is a need to tune machine dampers for a particular application to improve performance[46,47], but current simulation approaches fail to depict the typical hysteresis. Further, multi-material structures are plagued by stress concentrations during loading due to large stiffness gradients at the material boundaries[38]. Voxelated digital materials offer a potential solution since the damping material can be tuned for a target application and graded boundaries can be introduced between single, solid materials in the structure. The presented damper design is based on a Vibrostop AAT 20/N damper (Vibrostop SRL, Milan, Italy) with the interfaces between the compliant dampening material and the rigid housing showing a linear gradient to reduce stress concentrations. The mesoscopic material formulation enables the representation of the full structure on a 405,769-point grid instead of using $273.3 \cdot 10^6$ individual voxels. The discussed mesoscopic representation, the tetrahedral finite element mesh and the AM damper are shown in Fig. 5a. The viscoelastic material formulation in this work enables the prediction of the voxelated damper's hysteresis, which was not feasible in previous work on voxelated digital materials due to the typical adherence to linear elasticity and a quasi-static time scale[19]. The hysteresis is simulated and tested at three different frequencies (0.1, 1, and 10 Hz), each at two temperatures (15 °C and 25 °C). The simulation is performed in Abaqus CAE, with runtimes ranging from 100 to 145 min for a 253,777-element mesh assigned to 28 individual percolated digital material sections. The experimental and simulated force-displacement hysteresis curves are shown in Fig. 5b. The best agreement is observed at 0.1 Hz, with a mean reaction force (RF) error of just 5.6% across both temperatures. At 1.0 Hz, the mean error increases to 20.0% and at 10 Hz, it reaches 26.6%. The overall accuracy of the predicted hysteresis is significant, given the challenges

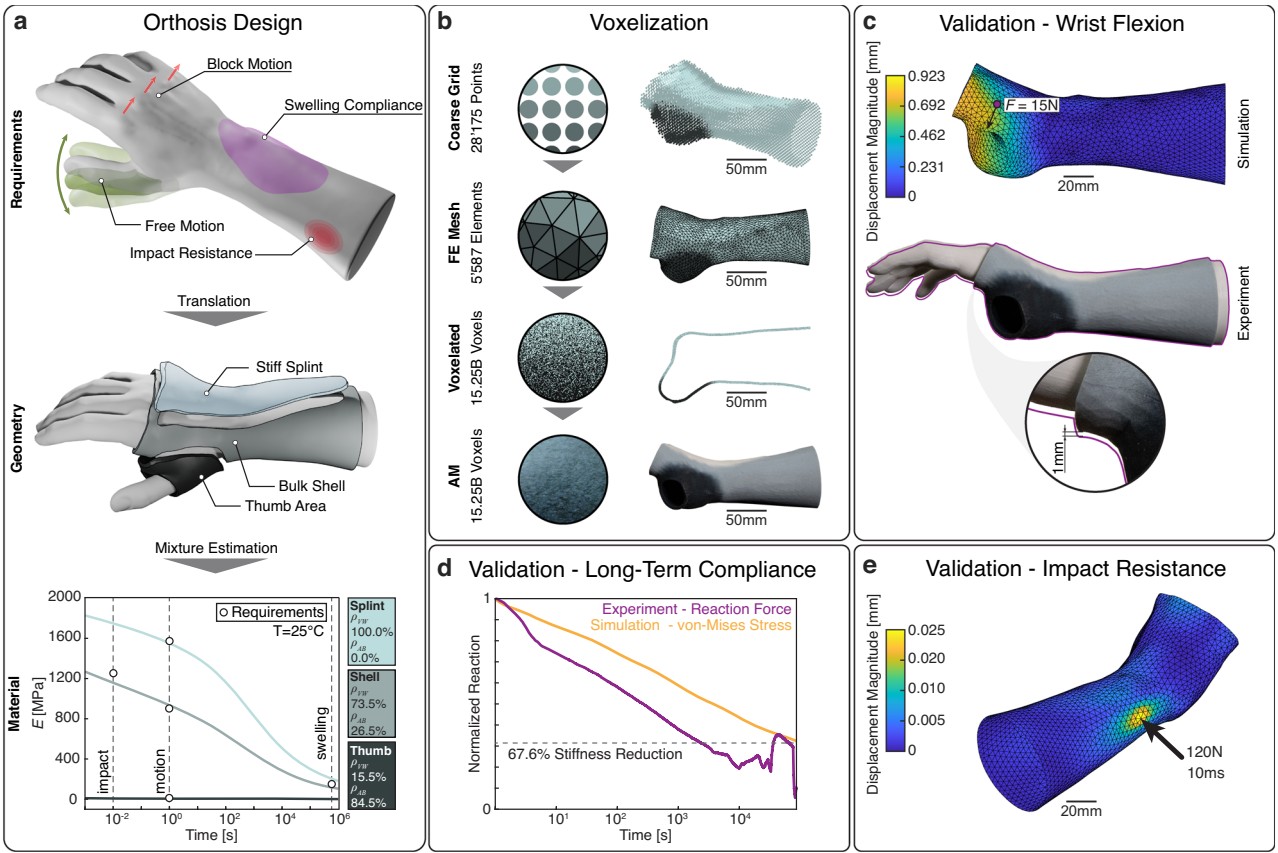

**Fig. 6 | Results related to the orthosis case study. a** Schematic requirements of the mechanical behavior of a static wrist orthosis and their translation into material zones in the orthosis geometry. The material requirement tuples are shown as circles in the stiffness-time plot. **b** Translation from mesoscale coarse grid to final printed structure with relevant entity counts. **c** Experimental and simulated wrist flexion and extension. The results of the wrist extension test are shown in the supplementary information. **d** Experimental and simulated swelling compliance of the shell material over a 24 h period. **e** Simulated impact resistance of the orthosis shell material. Source data are provided as a Source Data file. All images of the wrist are derived from a 3D scan provided by Theebadge/printables.com.

of simulating a complex 3D structure with graded material transitions subject to cyclic loading across multiple frequencies and temperatures.

## Case study: personalized wrist orthosis

The direct inverse design of the material distribution of a personalized, multi-material wrist orthosis is now shown. Personalization in healthcare often requires versatile design methods that can provide a design estimate for multiple local requirements, such as long-term compliance and short-term stability at the same location. While patient-specific design requirements vary significantly across patients and pathologies, and are not available in this study, the orthosis design is based on literature-derived design requirements to demonstrate the method's potential. The design requirements are formulated for three design regions all having a material thickness of 3 mm: the thumb region, the stabilization splint, and the bulk shell (Fig. 6a). The stabilization splint provides resistance against wrist flexion and extension allowing a maximum displacement of 2 mm under a force of 15N[48]. The thumb region must permit free thumb motion, and the bulk shell should provide impact resistance while still being compliant to long-term changes of the underlying anthropometry, such as swelling (Fig. 6a). The requirements for the stabilization splint and the thumb region are thus formulated as the maximum and minimal material stiffness available. The bulk shell is subject to more than one requirement, impacts are approximated as a concentrated force of 120 N over a time of 0.01 s, while the long-term material compliance must relieve a pathologic pressure of 150 mmHg to a physiological pressure below 25 mmHg within a day[49]. This translates to a stiffness of 1250 MPa at a

time scale of 0.01 s for impact resistance and a stiffness decrease from 900 MPa to 150 MPa within 86400 s (see supplementary information for further details).

The design requirements for the three orthosis regions are provided to the ANN and the resulting digital materials are used to generate an orthosis with distinct material regions and linear material gradients in-between. The coarse grid representation reduces a total of 15.25 billion voxels to 28,175 grid points. The proposed material model enables the simulation of orthosis flexion and extension, long-term compliance, and impact resistance in Abaqus CAE using a 5587-element mesh assigned to 19 individual percolated digital material sections. The wrist flexion and extension are simulated as a radially exerted force of 15 N and tested experimentally on a custom rig using an additively manufactured wrist and 15 N of weights attached to the palm. The more critical case of flexion shows an experimental and simulated displacement of 1 mm and 0.923 mm, respectively (Fig. 6c), indicating high accuracy considering the complex load case. The long-term compliance is experimentally evaluated by measuring the normalized RF of a half pipe of the shell material on an indenter initially loaded with 6 N over a period of 24 h. The simulation is performed by applying a radial displacement on the shell for 24 h. The normalized RF is compared to the normalized simulated reaction pressure and shows good correspondence (Fig. 6d). To further validate long-term stability, the shell material is tested under cyclic loading for 10,000 cycles, and no significant change in stiffness is observed (see supplementary information). The impact resistance is evaluated in simulation only with a ramped concentrated force to the shell (Fig. 6e) and shows very low displacement results, indicating that the orthosis can absorb

impacts while still exhibiting long-term compliance to account for swelling. Additional analysis of these four load cases at 0 °C (cold weather) and 37 °C (body temperature) is included in the supplementary information. The aggregated results across all three temperatures indicate that the multifaceted requirements placed on the orthosis have been fulfilled, both in the simulated and experimental behavior. The ability to predict complex time-dependent and temperature-dependent behavior in voxelated digital materials enabled by the presented modeling approaches is a crucial step towards highly personalized wearables.

## Discussion

The proposed material model is created and validated for mixtures of two materials using the polymer material jetting process. A key property of these two base materials is that they can both be modeled using a viscoelastic model and temperature shifting based on the WLF and Arrhenius function. However, the validity for other materials and AM processes needs to be addressed. Previous research shows percolation theory to be a valid model to approximate stochastic digital materials[19]. Results from Fig. 3 show that the percolation effect changes depending on the temperature and frequency. Since percolation effects can be related to the activation of free polymer chains[37], temperature and frequency are prone to influence the effect due to their influence on the chain motion[50]. This validates the use of a physics-based percolation model over empirical curve fitting, which is more commonly employed in voxel-based modeling[8,18]. Since these percolation effects can be observed in many types of polymers, the proposed approach is thought to be valid if the scale difference between material assignment and the voxel size allows the assumption of homogenized properties. Additionally, manufacturing inaccuracies shown in Fig. 2 cause complex intertwined interfaces between the individual voxels that are likely unique to the specific base materials and AM process. Expansion to other base materials would require a repetition of the experimental testing of the material mixtures and base materials. While previous methods used seven[8] to eleven[14,22] different digital materials to fit a model valid for a specific temperature and frequency, only five samples are used to fit the proposed model over the full temperature and frequency range. This is a key benefit compared to traditional homogenization methods and data-driven approaches since the data gathering techniques that allow high-value data extraction, such as DMA, are associated with high experimental effort. It is likely that the temperature and frequency dependency of the percolation coefficient varies across base material combinations, depending on the material interfaces and material entanglement. Further, the percolation threshold is thought to vary depending on the magnitude of smearing in the manufacturing process. In future research, it could potentially be possible to leverage the knowledge of the percolation coefficient variation and the percolation threshold for certain base materials and directly estimate the mixture behavior and temperature shifting by only knowing the base material behavior. This could even be expanded to more than two base materials, but needs intermediate testing to evaluate a multidimensional variation of the percolation coefficient.

The two presented case studies illustrate both the capabilities of the proposed modeling approach as well as the potential novel applications that are now made possible. For the first time, a general approach can be used to model viscoelastic behavior of voxelated digital materials over multiple time scales and temperatures with a high accuracy. The damper (Fig. 5) is a prime example of a system where predicting complex material behavior is of vital importance. The results show that it is possible to predict the cyclic behavior of a complex voxelated structure with graded material transitions with a 99.85% reduction in data points. This represents a significant improvement compared to the previous status quo of voxel-level linear elastic modeling in the quasistatic domain. The wrist orthosis

(Fig. 6) demonstrates that the presented modeling approaches leverage the potential of voxelated digital materials and gradients to personalize a medical device subject to complex requirements while simultaneously achieving a data point reduction by seven orders of magnitude. The ANN supports the designer in picking the correct material mixing ratios for all three zones of the orthosis by proposing materials that fit the specified requirements. For both case studies, FE simulations using the proposed material model accurately predict the behavior of the overall structure, closely matching the experimental results within a reasonable time frame due to the efficient mesoscale material representation. However, an increasing mean error at higher temperatures is observed for the damper (see Fig. 5b). This larger error is likely due to the limitations of the Maxwell model and shifting functions. The WLF and Arrhenius function are limited in their modeling of frequency-dependent shifting biases. In future work, a custom shifting function could be developed based on the output of the ANN to create an even more accurate material response in the FE simulation.

In contrast to the field of architected metamaterials, the proposed models do not architect the geometry by structural design but architect the material itself by determining an estimated mixed material for given requirements. Since the material model considers the material properties over multiple behavioral domains, it can be leveraged to tune stiffness, shape memory behavior, and dampening behavior. This provides a baseline for voxelated digital material assignment and unifies the currently existing design approaches by unifying the material behavior over all temperatures and time scales. As such, the presented approach is a task-independent inverse material design framework for viscoelastic stochastic digital materials. A valuable extension to this work could be the mapping of base material behavior to the homogenized properties of the voxelated digital material without testing intermediate mixtures. However, the stochastic nature of the voxel boundaries and the variability of the AM process make such an approach challenging, strengthening the need for a common rule of mixture as a homogenization approach, such as percolation.

While the proposed data representation achieves a data reduction over multiple orders of magnitude, we acknowledge several practical considerations. In terms of design, the computational bottleneck lies in generating the voxel images (see supplementary information). Although this is needed only once per design, the process could be sped up by more efficient algorithms. From a simulation perspective, the data reduction does not have an influence on the simulation accuracy because the simulation mesh is generated based on the coarse grid. The simulation will thus be accurate if the grid is smaller than the features to simulate. In terms of fabrication, voxel-level material placement offers high design versatility, but faces several industrial challenges. The use of dense support material increases waste and manufacturing time, especially for complex, internal geometries. Further, multi-material AM naturally causes higher production time and cost relative to single material processes. We thus envision the application of this framework in low-volume, personalized applications.

This work presents a modified, frequency-dependent percolation theory as a baseline modeling approach to accurately represent the time-dependent and frequency-dependent stiffness of stochastically mixed polymer digital materials consisting of two base components. This newly updated modeling approach is integrated into a loss-free, mesoscale material representation and an ANN is trained to inversely design voxelated digital materials based on given material property requirements. We show via experiments and simulations that the models can create tuned digital materials for complex loading, such as hysteresis and long-term relaxation, in seconds. These findings represent a significant step towards a generalized modeling approach to stochastic digital materials and enable the computational design of large, high resolution voxel structures with tuned behavior across

multiple time scales and temperature scales for a variety of applications in engineering and science.

## Methods

### Materials
All samples and designs are printed on a Stratasys J850 Prime (Stratasys ltd., Eden Prairie, MN, USA) using the two build materials Agilus30Black and VeroUltraWhite, SUP706 is used for support structures. The prints are printed using the Voxel Print™ utility included in the GrabCAD Print Pro™ Package. The designs are generated in MATLAB R2024a[51] and imported in GrabCAD as a set of PNG images, where each print material is encoded by a separate color. The J850 machine has a layer height of 27 μm for voxelated manufacturing and a resolution of 600dpi and 300dpi in x- and y-direction, respectively. To align the planar resolutions, the pixels in the imported images are doubled in x-direction to form quasi-voxels with a size of $84.66 \times 84.66 \times 27$ μm. The prints are postprocessed by removing the support material mechanically by hand, followed by cleaning the part at a water jet cleaning station.

### Mechanical testing
The two base materials and five stochastic digital materials are tested with a DMA and a tensile test. The sample geometry is a type IV dogbone geometry according to ASTM D638 with the dogbone ends printed in VeroUltraWhite, the core printed in the digital material under test and the transition between the two being a 2 mm long linear material gradient. The seven material mixtures are AB/VW mixtures with an AB content of 0, 16.5, 33.3, 50, 66.6, 83.5, and 100%. The DMA testing is performed on an Instron ElectroPuls E3000 (Instron, Norwood, MA, USA) using an Instron Dynacell 2527 load cell with a range of ±5 kN, Instron fatigue-rated mechanical wedge action tensile grips and an Instron 3119-600 series temperature-controlled chamber. Instron WaveMatrix is used to record all relevant data and to calculate the storage modulus $E'$, loss modulus $E''$ and dissipation factor $\tan(\delta)$ for a sinusoidal displacement input for frequencies between 1 Hz and 18 Hz with 1 Hz increments. The measurement temperatures are set for each sample to force a change from the glassy to the rubbery state. The sample is pre-strained to always be under tension and the pre-strain is increased for an increasing temperature to compensate for thermal expansion. An additional tensile test is performed at temperatures where the samples are in the rubbery state with a speed of 3 mm/min to ensure quasi-static conditions. The resulting force is measured, yielding the hyperelastic stress-strain data for all samples.

### Material model
The material is modeled using a generalized Maxwell model initially consisting of a single, linear Hookean element $k_0$ and $n$ Maxwell elements consisting of a linear Hookean element of stiffness $k_n$ and a Newtonian dashpot, or viscous element, with a relaxation time of $\tau_n$. As shown in Eq. (2), the relaxation time is modulated using a scaling factor $\alpha_T$ based on the WLF and Arrhenius equation.

$$\log_{10} \alpha_T = \begin{cases} C_3 \left( \frac{1}{T} - \frac{1}{T_g} \right) & if \ T < T_g \quad \text{Arrhenius} \\ \frac{-C_1(T-T_g)}{C_2+(T-T_g)} & if \ T \geq T_g \quad \text{WLF} \end{cases} \quad (2)$$

The glass transition temperature $T_g$ is derived from the observed peak in $\tan(\delta)$, the WLF parameters $C_1$ and $C_2$ are fixed to 17.44 and 51.6, respectively according to Ferry[40]. The Arrhenius parameter $C_3$ is determined based on the TTSP. The shifting procedure is conducted using a numerical optimization scheme based on a Nonuniform Pattern Search algorithm. The objective is to shift the individual frequency sweeps to create a master curve that is as smooth as possible by

minimizing the second derivative along its entire length. This is achieved by fitting a spline through the individual data points gathered during the mechanical testing procedure and numerically determining the second derivative. Thermal expansion is also considered using a thermal element with an expansion coefficient of $\alpha$. The experimental data is fit with a model consisting of $n = 31$ Maxwell elements, the estimated material in the design framework is also fit with a model with $n = 31$ Maxwell elements. The fitting procedure is conducted using a numerical optimization scheme based on a nonlinear least-squares algorithm. The fitted models are expressed using a normalized Prony series, the stiffness of the Maxwell elements is thus defined as a fraction of the long-term stiffness $k_0$. In the final simulation model, the Hookean element $k_0$ that models the long-term stiffness is replaced with a Mooney–Rivlin hyperelastic element. In combination with the normalized Prony series, this enables the simulation of larger strain ranges. The fitting of the Mooney–Rivlin hyperelastic model is conducted at runtime in the FE simulation software.

### Neural network setup
The neural networks $ANN_{mix}$ and $ANN_E$ are both trained on a dataset created using the temperature and frequency-dependent percolation. The stiffness $E$ is calculated for each temperature between $-30\,°C$ and $130\,°C$ with $1\,°C$ increments, each AB/VW mixture with 1% increments and each time between $10^{-15}$s and $10^{15}$s with 300 regular increments in the logarithmic space, yielding a dataset of 4.878 million data points. The training of $ANN_E$ is performed on the full data set and the training of $ANN_{mix}$ is performed on a reduced dataset, where all data for mixture ratios above the percolation threshold is culled to prevent one-to-many mapping. The full and reduced dataset are individually normalized to values between 0 and 1. The networks are trained on a training set with 80% of the data points and tested on a test set with the remaining 20%. Both networks are multi-layer perceptrons with a constant width and depth, an activation function globally defined for all neurons, an input size of three, and an output size of one. The network $ANN_{mix}$ has a width of 180, depth of seven (five hidden layers), GELU activation functions, and is trained using the ADAM optimization algorithm[52] with a learning rate of 0.001 and a batch size of 256 over 150 epochs. The loss function is the mean squared error, and the test loss of the trained network is around $10^{-5}$. The network $ANN_E$ has a width of 150, depth of seven (five hidden layers), GELU activation functions, and is trained using the ADAM optimization algorithm[52] with a learning rate of 0.0001 and a batch size of 256 over 150 epochs. The loss function is a weighted mean squared error with increased weights for lower stiffness labels to counteract the decreasing relative accuracy for labels around zero, a penalty term for stiffness estimations below zero is also added. The test loss after training is around $10^{-4}$. All hyperparameters are evaluated using a full factorial hyperparameter evaluation, including the ANNs width, depth, the activation function, learning rate, batch size, and number of training epochs (see supplementary information).

### Finite element simulation
All finite element simulations are performed in Abaqus CAE 2022 using a viscoelastic model with hyperelastic elements. The temperature shifting behavior is modeled using a custom UTRS user subroutine. The machine damper is simulated with linear 4-node tetrahedral elements (C3D4H) and the wrist orthosis is simulated with linear 3-node triangular shell elements of type S3R. The simulation details can be found in the supplementary information.

## Data availability
The processed material data, FE meshes and the ANNs generated in this study have been deposited in the GitHub repository (https://github.com/ETHZ-EDAC/STAMP) alongside the code. The raw experimental data is available under restricted access due to use in future

 

work, access can be obtained by contacting the corresponding author. Source data are provided with this paper.

## Code availability

The code[51] used in this study is available at https://github.com/ETHZ-EDAC/STAMP. The code is released under the MIT license under the following https://doi.org/10.5281/zenodo.17296598.

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

## Acknowledgements

Special thanks to Jan Weber for his assistance with the DMA measurements and Dr. Tino Stanković for his help in proofreading this manuscript.

## Author contributions

J.C. and M.W. contributed equally to the study and share the first authorship. J.C. developed the material model fitting procedure and the damper case study. M.W. developed the inverse design approach and the orthosis case study. J.C. and M.W. contributed equally to material testing, study conceptualization and writing of the original draft. K.S. contributed by project supervision, project administration, conceptualization, and reviewing and editing the manuscript.

## Funding

## Competing interests

The authors declare no competing interests.
