## [Transparent Peer Review file · Nature Communications]

Inverse Design of Stochastic, Voxeled Thermo-Viscoelastic Digital Materials

Corresponding Author: Mr Marc Wirth

Version 0:

Reviewer comments:

Reviewer #1

(Remarks to the Author)

This manuscript presents a computational framework for the inverse design of architected polymer structures with voxel-based microstructures, focusing on thermo-viscoelastic behavior. It combines percolation-inspired mixture modeling, Prony-series-based viscoelasticity, and a ANN model for inverse design. Applications to damping components and orthotic devices are included. While the manuscript is well written and the workflow is well implemented, the core methodology lacks novelty, and the proposed contributions seem incremental. Many similar frameworks already exist in the literature, and this work does not provide new theoretical insights, modeling techniques, or significantly more powerful design capabilities. Thus, the proposed work is not appropriate for publication in the journal Nature Communications, and a computation-focused journal would be more suitable.

1. The paper reuses well-established techniques—homogenized mesostructural modeling, time–temperature superposition, and ANN-based inverse design, without introducing significant innovation or theoretical advancement. In addition, the authors appear to miss the majority of the vast literature on the inverse design of architected structures and soft materials. There are a large number of studies that this manuscript ignores.
2. There is little to no quantitative comparison with existing frameworks for inverse design of soft architected materials, especially those using voxel representations and ML-based and gradient-based design tools. The authors should first conduct a comprehensive literature review and potentially revise and improve their approach.
3. There are two example applications (a damper and a wrist orthosis) that are rather simple and don't demonstrate the need for computational inverse design, and it is unclear how the method works for more technically challenging case (such as involving geometric constraints, multiple conflicting objectives, or large deformations).
4. The neural network is used as a black-box inverse predictor, but its architecture, training dynamics, generalization behavior, and failure cases are not well analyzed.
5. The ANN and mixture models assume deterministic relationships, but the actual AM processes and viscoelastic properties exhibit high variability.
6. The percolation-based mixture model is used, but not deeply analyzed or validated. It is unclear whether this offers any benefit over more traditional homogenization models in terms of prediction accuracy or computational cost.

Reviewer #2

(Remarks to the Author)

It is a great honor to receive an invitation from **[Redacted]** to review this manuscript from Nature Communications.

The overall recommendation is to accept it with minor revisions.

The detailed recommendation is as follows:

In this paper, an innovative reverse design framework, combining osmosis theory and neural networks, is proposed to provide a valuable tool for the design and simulation of multi-material voxel structures. It is very interesting. However, the academic rigor and engineering utility need to be further enhanced by additional experimental validation, quantifying the effect of fabrication errors, and refining the biomechanical assumptions. I recommend this manuscript to be accepted after a slight revisions.

a. The current model is based on a blend of two polymers (Agilus30Black and VeroUltraWhite) and the permeation threshold ($\mu_c=0.15$ $\mu_c=0.15$), but independent experimental validation against the material combinations of the present study is not provided. Furthermore, whether the frequency dependence of the permeation index $\mu\mu$ is applicable to other material systems (e.g., thermoplastic elastomers versus rigid plastics) has not been discussed.

Recommendation:

- Add experimental validation of permeation thresholds (e.g., microstructural characterization) for the selected material combinations.
- Discuss the applicability of the model to other material systems, or clarify the scope of application in the "Limitations" section.

b. The training data is generated by the theoretical model (1.6M data points) and not directly based on experimental data. Although the theoretical model has been partially validated, it may result in a network that is not robust to actual manufacturing errors (e.g., material "fuzziness").

Recommendation:

- Introduce test data from actual print samples as part of the training set to enhance the generalization of the model.
- Provide a sensitivity analysis of the network under noisy data (e.g., prediction error after randomly perturbing the mixing ratio).

c. The design requirements for the wrist orthosis (e.g., maximum displacement of 2mm at 15N force) are based on literature 2929 but do not incorporate actual biomechanical data from the patient (e.g., joint stiffness for different pathologies). Fatigue properties for long-term use (e.g., material relaxation after cyclic loading) were also not validated.

Recommendation:

- Supplement patient-specific biomechanical data or cite broader clinical studies to support design criteria.
- Add long-term fatigue testing (e.g., change in stiffness after 10,000 cycles of loading).

d. The paper points out the "blurring" phenomenon during material injection (Figure 2a), but does not analyze its impact on simulation accuracy. For example, the actual printed gradient transition region may deviate from the linear mixing assumption of the design.

Recommendation:

- Quantify manufacturing errors (e.g., analyze actual material distribution by micro-CT) and introduce tolerance parameters in the model.
- Discuss the extent to which manufacturing errors affect case study performance (e.g. damper hysteresis curves).

e. Simulation parameters (e.g., mesh sensitivity analysis, time step selection) are not specified, and no mention is made of the convergence of multi-threaded calculations (the supplementary material mentions a single-threaded Abaqus run). In addition, the parameter fitting procedure for the hyperelastic model (Mooney-Rivlin) is not disclosed.

Recommendation:

- Add grid-independent validation (e.g., comparison of displacement errors at different grid densities).
- Disclosure of the fitting method and experimental data support for the parameters of the hyperelastic model.

f. The paper emphasizes the "high efficiency" of the method (7 orders of magnitude reduction in data points) but does not discuss the practical limitations in its engineering applications, e.g.:

- The demand on computational resources for high-resolution voxel design (e.g., preprocessing time for 280,000 grid points).
- Cost and scalability of multi-material printing (e.g., difficulty in removing the support material SUP706).

Recommendation:

- Add an analysis of the trade-off between computational efficiency and accuracy (e.g., mesh resolution vs. simulation error).
- Discuss economic challenges in industrial-scale applications (e.g., material waste rate, print time).

All the best to **[Redacted]** and everyone! Have a good day! Yuze Nian May 1, 2025

Reviewer #3

(Remarks to the Author)

This manuscript presents a novel inverse design framework for voxelated thermo-viscoelastic materials. This approach effectively reduces computational complexity via mesoscale homogenization and demonstrates promising predictive accuracy in two functional case studies. While this work offers a valuable contribution to data-driven design of architected soft materials, its limited validation is a significant weakness. Therefore, I would like to recommend it for publication after addressing some concerns. Detailed comments are provided below.

- The manuscript relies heavily on simulation results without providing adequate experimental validation, which may raise questions about the practical applicability of the findings. The Prony series fitting is based on only four experimental datasets for training and a single dataset for validation, which may be insufficient to robustly capture the full range of material

behavior, as acknowledged by the authors. Moreover, the inverse design framework relies solely on simulations. If possible, incorporating such experimental validation would significantly enhance the credibility of the approach.

- Although the authors emphasize the importance of temperature-sensitive material responses, most of the work are conducted at room temperature. Extending the analyses to include elevated or variable temperatures, especially those relevant to the case studies, would better demonstrate the robustness and applicability of the model.
- I would recommend replacing the term “graduated” with “graded” to reduce potential confusion among interdisciplinary readers, as it is the more commonly accepted term in the related field.
- Page 7, line 186: please check the error.

Reviewer #4

(Remarks to the Author)

Version 1:

Reviewer comments:

Reviewer #1

(Remarks to the Author)

The authors have revised according to the reviewer's comments and the revised manuscript is improved. Thus I have no other comments for the paper.

(Remarks on code availability)

Reviewer #2

(Remarks to the Author)

Cheers! All my comments have been responded to. The paper contains useful results and is ready for publication. Congratulations and looking forward to further research results! It is a great honor to receive an invitation from **[Redacted]** to review this manuscript from Nature Communications. Best, Yuze Nian. August 12, 2025.

(Remarks on code availability)

Based on my current capabilities, the code in this paper is highly feasible, reproducible, and accurate, and can support the publication of this paper. Best, Yuze Nian.

Reviewer #3

(Remarks to the Author)

The authors have revised the manuscript in response to the initial review, addressing several of the previously raised concerns. The integration of experimental data to validate the simulation results of the storage modulus of voxelated materials under frequency- and temperature-dependent conditions enhances the credibility of the proposed approach. The inclusion of multiple temperature conditions in the case study analyses is appropriate, as it highlights the method's ability to account for different loading scenarios. Changing the term graduated to graded has also minimized potential confusion for readers, aligning the terminology with common usage in the field. Overall, the manuscript has improved meaningfully, and the authors have been responsive to reviewers feedback.

That said, I have a few remaining or new points for consideration before recommending acceptance:

- After the mixture ratio is determined, it would be helpful to clearly describe how the mesoscale unit size (coarse grid size) for applying the stochastic distribution is selected in practice. Providing explicit criteria or a decision process for choosing the size and shape of these units, preferably in a way that allows readers to visualize the workflow, would help future researchers replicate the proposed method.
- In the damper experiment, the error increased to 26.6% under the 10 Hz condition. It would strengthen the manuscript to include an analysis of the cause of this reduced accuracy in the high-frequency range and to discuss potential improvement strategies (e.g., augmenting ANN training data or adjusting model parameters).
- In Figure 2a, the term graduated is still used. Please replace it with graded for consistency with the rest of the manuscript.

(Remarks on code availability)

Reviewer #4

(Remarks to the Author)

(Remarks on code availability)

The repository provides clear and sufficient instructions, and I was able to run the code without major issues. The results presented in the paper are reproducible using the provided implementation.

Response to the Reviewers' Comments for Manuscript:
"Inverse Design of Stochastic, Voxelated Thermo-Viscoelastic Digital
Materials"

Dear Reviewers,

Thank you very much for the helpful comments to further improve the quality of our manuscript. We have carefully considered every comment and address them here, in addition to the changes made in the revised manuscript that are marked using a yellow highlight and the corresponding identifier [R*C*]. Following your comments, we conducted extensive additional experimental validation and literature review. This additional validation led to refitting the material model and complete retraining of the ANN to include the newly generated experimental data. The main paper has been updated with the new results, and the supplementary information has been significantly expanded to include additional analysis and results. To better emphasize the scope of the work, the title has been changed to "Inverse Design of Stochastic, Voxelated Thermo-Viscoelastic Digital Materials".

We further made additional minor changes to the manuscript to improve the readability and clarity.

The Authors

Reviewer #1: *This manuscript presents a computational framework for the inverse design of architected polymer structures with voxel-based microstructures, focusing on thermo-viscoelastic behavior. It combines percolation-inspired mixture modeling, Prony-series-based viscoelasticity, and a ANN model for inverse design. Applications to damping components and orthotic devices are included. While the manuscript is well written and the workflow is well implemented, the core methodology lacks novelty, and the proposed contributions seem incremental. Many similar frameworks already exist in the literature, and this work does not provide new theoretical insights, modeling techniques, or significantly more powerful design capabilities. Thus, the proposed work is not appropriate for publication in the journal Nature Communications, and a computation-focused journal would be more suitable.*

R1C1: *The paper reuses well-established techniques—homogenized mesostructural modeling, time–temperature superposition, and ANN-based inverse design, without introducing significant innovation or theoretical advancement. In addition, the authors appear to miss the majority of the vast literature on the inverse design of architected structures and soft materials. There are a large number of studies that this manuscript ignores.*

An additional literature review has been conducted, and the introduction has been significantly rewritten. The key novelty of the presented work is the material modeling of high-resolution voxelated digital materials that are geometry-independent and use physics-based percolation material modeling. Specifically, the novelty lies in the definition of a time-dependant and temperature-dependent adaption of the de Gennes percolation theory. This modeling approach enables the simulation of complex behavior such as hysteresis, shape memory effect, and stiffness modulation across a wide range of time scales and temperatures. Additionally, the combination of the efficient mesoscale representation and the ability to predict the properties of any digital material composed of the two base materials enables the simulation of complex graded transitions in a computationally efficient manner while undergoing complex loading. The experimentally validated ANN-assisted inverse digital material design method enables designers to determine a digital material that satisfies given requirements at various temperatures and time scales. In summary, these findings represent a significant step towards a generalized modeling approach to stochastic voxelated digital materials and enable the computational design of high resolution voxelated structures with tuned behavior across multiple time and temperature scales for a variety of applications in engineering and science.

R1C2: *There is little to no quantitative comparison with existing frameworks for inverse design of soft architected materials, especially those using voxel representations and ML-based and gradient-based design tools. The authors should first conduct a comprehensive literature review and potentially revise and improve their approach.*

Many existing inverse design frameworks in the field of metamaterials and soft materials focus on inverse geometric design, such as optimizing unit cell topologies and lattice layouts. In contrast, our framework determines digital material allocation for a given geometry. Thus, we changed the terminology in the paper to “Inverse Design of Stochastic, Voxelated Thermo-Viscoelastic Digital Materials”. The challenge addressed in this paper lies in creating stochastic voxelated digital materials to meet viscoelastic, time-dependent and temperature-dependent requirements. The goal of our work is not to generate local shapes or structures, but rather to determine a fitting digital material that satisfies the user requirements. This

makes existing geometric optimization frameworks not directly comparable to the problem we address, but rather complementary. To emphasize this distinction and to highlight the contributions of this work, we extended the introduction and related this work more tightly to the existing body of research. We hope this clarification resolves the concern and helps to position our unique contribution.

R1C3: *There are two example applications (a damper and a wrist orthosis) that are rather simple and don't demonstrate the need for computational inverse design, and it is unclear how the method works for more technically challenging case (such as involving geometric constraints, multiple conflicting objectives, or large deformations).*

The damper and orthosis case study have been adapted to better show the unique capabilities of the model. The damper is now validated at two different temperatures with loading at three different frequencies for each temperature. The orthosis is validated at two additional temperatures. These additional results have been added to Fig. 5 and Fig. 6 in the main text and are further discussed in the supplementary information.

The context of the inverse digital material design approach has been further clarified as described in [R1C2]. Since the goal of our work is not to generate local shapes or structures, but rather to determine the digital material composition that satisfies the user requirements, geometric constraints cannot be addressed directly. In the case of conflicting material requirements, the current method determines a digital material that represents a tradeoff between the different objectives. The inverse digital material design method uses the dataset of all percolation-based material master curves to determine the mechanical response of the stochastic digital material. When there is large deformation or complex, nonlinear load cases, the global mechanical response of the macroscale structure is defined by a strong interplay between the digital material composition, the geometry of the structure and the load history. To address this interplay, the proposed modeling approach could be integrated into a numerical optimization scheme that controls coarse grid digital material assignment to find an optimal digital material distribution for the given global load case. This is outside the scope of the presented work.

R1C4: *The neural network is used as a black-box inverse predictor, but its architecture, training dynamics, generalization behavior, and failure cases are not well analyzed.*

We have added a detailed analysis of the ANNs hyperparameter selection, training behavior, and ANN accuracy to the supplementary information (section C, Figure S14-S17). The hyperparameter selection is performed by a full factorial analysis and results in ANNs with lower training loss and test loss. For the ANNs with the lowest loss, the convergence of the tests show that the data is not overfit and the ANN acts as a good surrogate for the underlying dataset. Further, we provide accuracy maps to visualize the spatial accuracy of the ANN to the test data. The accuracy is interpreted and set into the context of this work. The ANN estimations show a high accuracy over the full range of all parameters, this demonstrates that the ANNs operate as transparent, well-posed interpolators, rather than a black-box predictor. We further show that the failure cases, i.e. where the ANN matches the dataset the least, occur in regions with limited mechanical sensitivity and have negligible impact on the mechanical properties of the resulting digital material.

R1C5: *The ANN and mixture models assume deterministic relationships, but the actual AM processes and viscoelastic properties exhibit high variability.*

We fully agree that the experimental viscoelastic behavior of AM materials can be affected by both the variability of the AM process and variability of the stochastic material assignment. We added a quantification of both variabilities by a combined experimental and analytical analysis, where details are given in the supplementary information (Section A.4) and summarized in the main text. Manufacturing variability is characterized by uniaxially testing 20 gradated dogbone samples with the same stochastic seed. We show that the manufacturing variability is 3% at peak reaction force. Variability due to stochastic distribution is experimentally characterized by uniaxially testing 20 gradated dogbone samples with unique stochastic seeds but having the same mesoscale material allocation. These show a variability of 2.5%, which is below the manufacturing variability. We further extend the analysis of the design variability by an analytical approach, where we show that stochastic material allocation variability only becomes significant for features sizes below 1-2mm. This supports the validity of the deterministic modeling approach for the mesoscale structures targeted in this work.

R1C6: *The percolation-based mixture model is used, but not deeply analyzed or validated. It is unclear whether this offers any benefit over more traditional homogenization models in terms of prediction accuracy or computational cost.*

The analysis of the percolation-based mixture model has been significantly expanded. Two additional digital materials have been tested experimentally. The model is now fit in a two-stage approach based on five digital materials and is subsequently validated with two digital materials instead of one (Fig. 3d). The percolation exponent that results from the fitting procedure is also analyzed in more detail in Fig. 3c and the corresponding text. The key benefit of the presented approach over more traditional homogenization models is the relatively low number of samples needed to conduct the material fitting procedure due to the use of a physics-based percolation model. While previous methods use seven¹ to eleven^{2,3} different digital materials to fit a model valid for a specific temperature and frequency, we use only five to fit a model over the full temperature and frequency range. The low sample number is especially significant since the data gathering techniques that allow high-value data extraction, such as DMA, are associated with high experimental effort.

Reviewer #2: *In this paper, an innovative reverse design framework, combining osmosis theory and neural networks, is proposed to provide a valuable tool for the design and simulation of multi-material voxel structures. It is very interesting. However, the academic rigor and engineering utility need to be further enhanced by additional experimental validation, quantifying the effect of fabrication errors, and refining the biomechanical assumptions. I recommend this manuscript to be accepted after a slight revisions.*

R2C1: *The current model is based on a blend of two polymers (Agilus30Black and VeroUltraWhite) and the permeation threshold ($\rho_c=0.15$ $\rho_c=0.15$), but independent experimental validation against the material combinations of the present study is not provided. Furthermore, whether the frequency dependence of the permeation index μ is applicable to other material systems (e.g., thermoplastic elastomers versus rigid plastics) has not been discussed.*

Recommendation:

-Add experimental validation of permeation thresholds (e.g., microstructural characterization) for the selected material combinations.

-Discuss the applicability of the model to other material systems, or clarify the scope of application in the “Limitations” section.

This comment addresses an important point regarding both experimental validation and model generalizability. To address the experimental validation, we have added an additional validation of the percolation threshold for the Agilus30Black (AB) and VeroUltraWhite (VW) system used in this study (Supplementary Information A.3). In this qualitative validation, a set of digital material samples with varying base material ratio is fabricated and visually inspected, confirming that the transition from connected to dispersed material regions occurs between 15 and 20% AB content. Further, we have characterized two further digital material samples using DMA testing to validate the percolation-based material model. The two new digital material samples have base material mixture ratios of 16.5% AB and 83.5% VW and vice versa. These two digital material samples and the previous validation sample with a 50/50 AB/VW base material mixture are placed between the four initial samples to set up the model. The percolation threshold ρ_c is now determined based on the 16.5% VW and 83.5% AB sample using an additional fitting step on top of the fitting of the percolation coefficient μ . The accuracy of the model prediction to the validation sample data of two samples, as well as the accuracy of the model prediction to the samples used for fitting, shows that the proposed model is valid across the whole design domain.

To clarify the generalizability of the frequency-dependent percolation coefficient across different material systems, we have expanded the discussion to explicitly state that the temperature and frequency dependence may vary with different material systems. This variation is likely influenced by interfacial interactions, polymer chain entanglement, and material smearing during fabrication. We thus argue that applying the model to other base material combinations would require a new fitting procedure that requires only five samples (2 base materials and 3 digital materials) to be tested.

R2C2: *The training data is generated by the theoretical model (1.6M data points) and not directly based on experimental data. Although the theoretical model has been partially validated, it may result in a network that is not robust to actual manufacturing errors (e.g., material “fuzziness”).*

Recommendation:

- Introduce test data from actual print samples as part of the training set to enhance the generalization of the model.
- Provide a sensitivity analysis of the network under noisy data (e.g., prediction error after randomly perturbing the mixing ratio).

While the ANN is trained on the theoretical model data, we have taken steps to assess its robustness in realistic settings. We analytically quantified the standard deviation in effective mixture ratio of the base materials due to the stochastic base material assignment and evaluated the expected difference in estimated stiffness in Section A.4 of the supplementary information. This quantifies the stiffness misestimation compared to experimental conditions and also with respect to feature size. The ANN is trained on model data since the experimental data from five samples leaves significant data gaps between the measured material mixes. The model data thus fills these gaps by physically modeling the behavior with the de Gennes percolation theory. Inclusion of raw experimental data is avoided since mixing model and experimental data leads to a noisier dataset and duplicate entries. This will effectively lead to worse ANN training.

R2C3: *The design requirements for the wrist orthosis (e.g., maximum displacement of 2mm at 15N force) are based on literature 2929 but do not incorporate actual biomechanical data from the patient (e.g., joint stiffness for different pathologies). Fatigue properties for long-term use (e.g., material relaxation after cyclic loading) were also not validated.*

Recommendation:

- Supplement patient-specific biomechanical data or cite broader clinical studies to support design criteria.
- Add long-term fatigue testing (e.g., change in stiffness after 10,000 cycles of loading).

A current state-of-the-art wrist orthosis design is either manually manufactured as a plaster cast or additively manufacturing as a rigid orthosis. The stiffness of the orthosis is typically tuned heuristically on a patient-by-patient basis. To our knowledge, patient-specific biomechanical data does not currently exist in the public domain and has not been quantified in a generalized approach in literature. The design requirements in this paper are derived from the body of available literature. This assumption and the choice of design variables is clarified in the revised manuscript. We hope that this clarification highlights our focus on validating the design and modeling frameworks under generalizable conditions, while future work may incorporate patient-specific data when available.

To address the concern regarding long-term material durability, additional experimental validation of fatigue performance under cycling loading is added to the supplementary information (section E.5). A standardized dogbone sample made from orthosis shell material is tested and showed no significant reduction in stiffness after 10'000 cycles of loading. This demonstrates the materials' suitability for long-term cyclic applications, such as orthoses.

R2C4: *The paper points out the “blurring” phenomenon during material injection (Figure 2a), but does not analyze its impact on simulation accuracy. For example, the actual printed gradient transition region may deviate from the linear mixing assumption of the design.*

Recommendation:

- Quantify manufacturing errors (e.g., analyze actual material distribution by micro-CT) and introduce tolerance parameters in the model.

- *Discuss the extent to which manufacturing errors affect case study performance (e.g. damper hysteresis curves).*

We have added both experimental and analytical analyses to quantify the effect of blurring, stochasticity, and manufacturing variability. Detailed analysis is shown in the supplementary information (A.3 and A.4). The findings are also summarized in the main text.

First, we analyze the additively manufactured morphology of stochastic digital material across varying base material ratios. A visual inspection confirms a clear transition from connected to dispersed regions between 15% and 20% VW content. This partially validates the literature-derived assumption of the percolation threshold. A new dogbone sample at 16.5% VW content is added to the percolation fitting procedure to more accurately model the critical percolation threshold (Fig. 3). Further, it provides an empirical understanding of how base material transitions evolve during the additive manufacturing process, including blurred boundaries between regions.

Second, we conduct a repeatability study, where we distinguish between two sources of variability: manufacturing process variation and stochastic voxel layout. Mechanical testing of 20 samples with identical voxel layout shows a manufacturing-induced variability of 3% in reaction force. Characterization of 20 samples with randomized voxel layouts, but with the same base material mix ratio, shows variability of 2.5%. This shows that blurring and stochasticity have limited effects for feature sizes of the tested magnitude relative to manufacturing accuracy.

To further assess the influence of stochasticity at smaller scales, we conduct an analytical study. We model the stochastic assignment as a binomial process and quantify the standard deviation in the effective mixture ratio relative to the target base material mix ratio as a function of feature size. We propagate this error through the ANN to evaluate the mismatch in estimated material stiffness. We deduce that the stiffness error increases significantly for sub-millimeter features and shadows the manufacturing variability of 3% for features below 1-2mm, depending on the base material mix ratio.

While quantification of actual base material distributions within a digital material with 3D visualization methods is an excellent next step, our current combination of morphological inspection, experimental testing, and analytical modeling provides a validated feature size threshold for digital material assignment. Importantly, all features in the damper and orthosis case studies are above the critical threshold of 1-2mm, ensuring that base material allocation variability does not critically impact the case study results.

R2C5: Simulation parameters (e.g., mesh sensitivity analysis, time step selection) are not specified, and no mention is made of the convergence of multi-threaded calculations (the supplementary material mentions a single-threaded Abaqus run). In addition, the parameter fitting procedure for the hyperelastic model (Mooney-Rivlin) is not disclosed.

Recommendation:

- *Add grid-independent validation (e.g., comparison of displacement errors at different grid densities).*
- *Disclosure of the fitting method and experimental data support for the parameters of the hyperelastic model.*

Additional simulation parameters are added to the Supplementary Information for both case studies. As mentioned in the Supplementary Information, multi-threaded runs using the Abaqus viscoelastic step show significant convergence issues. We observe that models exhibiting stable convergence in a single-threaded environment often failed to converge under multi-threaded execution even when subject to significant time step reduction or relaxed viscoelastic strain error tolerances. This could also be caused by the custom subroutine used to model Arrhenius and WLF behavior. However, to the best of our knowledge, the current version of the subroutine is thread safe.

A specific mesh convergence study is not conducted as there is an interaction between feature size, coarse grid spacing, and element sizing in the FE simulation. The coarse grid needs to be chosen such that it can accurately assign material at the minimum feature size of the current structure. Material is assigned to the FE mesh based on the materials assigned to coarse grid points. The sizing of the FE mesh is thus directly related to the sizing of the coarse grid. A full mesh sensitivity analysis exceeds the scope of this work as the focus of this work is mainly on the material model and less on the specifics of the FE model.

The Mooney-Rivlin hyperelastic model is fit at runtime by the Abaqus solver. The hyperelastic data is passed to Abaqus in the form shown in Table S2 with the setting “uniaxial test data”. The hyperelastic curve of the material mixtures is determined based on the hyperelastic curves of the two base materials (AB/VW in Table S2) and a mixture ratio based on the long-term response of the percolated master curve relative to the long-term response of the base materials. The online methods section has been adapted to better reflect this step.

R2C6: *The paper emphasizes the “high efficiency” of the method (7 orders of magnitude reduction in data points) but does not discuss the practical limitations in its engineering applications, e.g:*

- *The demand on computational resources for high-resolution voxel design (e.g., preprocessing time for 280,000 grid points).*
- *Cost and scalability of multi-material printing (e.g., difficulty in removing the support material SUP706).*

Recommendation:

- *Add an analysis of the trade-off between computational efficiency and accuracy (e.g., mesh resolution vs. simulation error).*
- *Discuss economic challenges in industrial-scale applications (e.g., material waste rate, print time).*

We added a paragraph to the discussion containing an assessment of the trade-offs involved. We now clarify the computational bottleneck and how to address it. Together with the response to [R2C4], where we analyzed feature size influencing manufacturing accuracy, we provide some practical guidelines for voxel-based design. On the manufacturing side, we have placed the proposed framework in the current additive manufacturing environment and provided recommendations on when to use it. Further, we added a section (B.11, including Fig. S13) to the supplementary information that shows the runtime for generating the designs of both case studies. The digital material

assignment on the coarse grid is very fast. This shows the framework's capability for fast design iterations.

Reviewer #3: *This manuscript presents a novel inverse design framework for voxelated thermo-viscoelastic materials. This approach effectively reduces computational complexity via mesoscale homogenization and demonstrates promising predictive accuracy in two functional case studies. While this work offers a valuable contribution to data-driven design of architected soft materials, its limited validation is a significant weakness. Therefore, I would like to recommend it for publication after addressing some concerns. Detailed comments are provided below.*

R3C1: *The manuscript relies heavily on simulation results without providing adequate experimental validation, which may raise questions about the practical applicability of the findings. The Prony series fitting is based on only four experimental datasets for training and a single dataset for validation, which may be insufficient to robustly capture the full range of material behavior, as acknowledged by the authors. Moreover, the inverse design framework relies solely on simulations. If possible, incorporating such experimental validation would significantly enhance the credibility of the approach.*

Additional experimental validation samples have been measured and incorporated into both the fitting and validation procedures. Fig. 3 has been reworked to include these new samples. The percolated material model is now compared to five digital materials (Fig. 3d), two of these digital materials are used to fit the percolation coefficient μ , one is used only for the fitting of the critical percolation threshold ρ_c , and two are used only for validation purposes. The entire material response is now split into six equally spaced bands by the two base materials and five stochastic digital materials. The dataset used to train both ANNs is generated from this fitted percolation model and is based on experimental data and validated by the samples not included in the fitting procedure.

Additionally, the damper case study in Fig. 5 has been expanded to include testing at multiple temperatures and frequencies, thus further validating the model and simulation approach based on the ANNs. The orthosis case study has also had additional simulations introduced at temperatures relevant to its application.

R3C2: *Although the authors emphasize the importance of temperature-sensitive material responses, most of the work are conducted at room temperature. Extending the analyses to include elevated or variable temperatures, especially those relevant to the case studies, would better demonstrate the robustness and applicability of the model.*

The damper case study and orthosis study have been extended to better show the capabilities of the model. The damper is now validated at two different temperatures (15°C and 45°C) with loading at three different frequencies (0.1Hz, 1Hz, 10Hz) for each temperature. This shows the predictive capabilities of the modeling approach for the complex load case of hysteresis. The orthosis is also validated at expected ambient temperatures and skin temperature. A cold environment simulation at 0°C is conducted, and a skin temperature validation at 37°C is also performed.

R3C3: *I would recommend replacing the term “graduated” with “graded” to reduce potential confusion among interdisciplinary readers, as it is the more commonly accepted term in the related field.*

We agree on using “graded” to enhance clarity for a broader audience. All terms have been adjusted accordingly in the manuscript and the supplementary information.

R3C4: Page 7, line 186: please check the error.

The error is a reference with the wrong hyperlink. We corrected it accordingly.

References

1. Saldívar, M. C., Doubrovski, E. L., Mirzaali, M. J. & Zadpoor, A. A. Nonlinear coarse-graining models for 3D printed multi-material biomimetic composites. *Addit Manuf* **58**, 103062 (2022).
2. Dong, L. & Wang, D. Optimal Design of Three-Dimensional Voxel Printed Multimaterial Lattice Metamaterials via Machine Learning and Evolutionary Algorithm. *Phys Rev Appl* **18**, 054050 (2022).
3. Yuan, C., Wang, F., Rosen, D. W. & Ge, Q. Voxel design of additively manufactured digital material with customized thermomechanical properties. *Mater Des* **197**, 109205 (2021).

Response to the Reviewers' Comments for Manuscript:
"Inverse Design of Stochastic, Voxelated Thermo-Viscoelastic Digital
Materials"

Dear Reviewers,

Thank you very much for the helpful comments to further improve the quality of our manuscript. We have carefully considered every comment and address them here.

We further made additional minor changes to the manuscript to improve the readability and clarity based on editorial input.

The Authors

Reviewer #1: *The authors have revised according to the reviewer's comments and the revised manuscript is improved. Thus I have no other comments for the paper.*

Reviewer #2: *Cheers! All my comments have been responded to. The paper contains useful results and is ready for publication. Congratulations and looking forward to further research results! It is a great honor to receive an invitation from [Redacted] to review this manuscript from Nature Communications.*

Based on my current capabilities, the code in this paper is highly feasible, reproducible, and accurate, and can support the publication of this paper.

Reviewer #3: *The authors have revised the manuscript in response to the initial review, addressing several of the previously raised concerns. The integration of experimental data to validate the simulation results of the storage modulus of voxelated materials under frequency- and temperature-dependent conditions enhances the credibility of the proposed approach. The inclusion of multiple temperature conditions in the case study analyses is appropriate, as it highlights the method's ability to account for different loading scenarios. Changing the term graduated to graded has also minimized potential confusion for readers, aligning the terminology with common usage in the field. Overall, the manuscript has improved meaningfully, and the authors have been responsive to reviewers feedback.*

That said, I have a few remaining or new points for consideration before recommending acceptance:

R3C1: *After the mixture ratio is determined, it would be helpful to clearly describe how the mesoscale unit size (coarse grid size) for applying the stochastic distribution is selected in practice. Providing explicit criteria or a decision process for choosing the size and shape of these units, preferably in a way that allows readers to visualize the workflow, would help future researchers replicate the proposed method.*

The corresponding description in the main manuscript has been rewritten to better highlight how the coarse grid resolution is chosen. In practice, the coarse grid resolution is adapted to the size of the geometric features in the structure, the desired material assignment resolution and the FE mesh resolution required for simulation.

Additionally, Section B.4 in the supplementary information has been expanded to discuss these three key considerations in more detail.

R3C2: *In the damper experiment, the error increased to 26.6% under the 10 Hz condition. It would strengthen the manuscript to include an analysis of the cause of this reduced accuracy in the high-frequency range and to discuss potential improvement strategies (e.g., augmenting ANN training data or adjusting model parameters).*

The discussion has been expanded to include additional information on what drives this error and what measure could be used to address it.

At higher frequencies the materials are generally in a region of larger stiffness gradients. This means that small inaccuracies in the material model fitting procedure and specifically temperature shifting can cause a more significant mismatch between the experiment and the

simulation. The results at 10 Hz and 25°C are also less accurate than the results at 15°C. This is driven by the limitations of the Maxwell model and shifting functions. The WLF and Arrhenius function are limited in their modeling of frequency-dependent shifting biases. The ANN response can model more complex behavior in terms of temperature shifting than the final Maxwell model as shown in Fig. 4f. A possible solution that could be used to address this issue is the use of a custom temperature shifting function for the Maxwell model based on the percolated material data.

R3C3: *In Figure 2a, the term graduated is still used. Please replace it with graded for consistency with the rest of the manuscript.*

Figure 2a has been updated to the new terminology.

Reviewer #4: *I co-reviewed this manuscript with one of the reviewers who provided the listed reports. This is part of the Nature Communications initiative to facilitate training in peer review and to provide appropriate recognition for Early Career Researchers who co-review manuscripts.*

The repository provides clear and sufficient instructions, and I was able to run the code without major issues. The results presented in the paper are reproducible using the provided implementation.